# The Problem of Antimalarial-Drug Abuse by the Inhabitants of Ghana

**DOI:** 10.3390/medicina59020257

**Published:** 2023-01-29

**Authors:** Ewa Zieliński, Marek Kowalczyk, Karolina Osowiecka, Łukasz Klepacki, Łukasz Dyśko, Katarzyna Wojtysiak

**Affiliations:** 1Department of Emergency Medicine, Collegium Medicum in Bydgoszcz, Nicolaus Copernicus University in Torun, 85-067 Torun, Poland; 2Department of Psychology and Sociology of Health and Public Health, University of Warmia and Mazury, 10-719 Olsztyn, Poland; 3Clinic of Oncological and General Surgery, University Clinical Hospital in Olsztyn, 11-041 Olsztyn, Poland; 4Department of Anatomy, University of Warmia and Mazury in Olsztyn, 11-041 Olsztyn, Poland; 5Department of Social and Medical Sciences, Faculty of Health Sciences, Collegium Medicum in Bydgoszcz, Nicolaus Copernicus University in Torun, 85-067 Torun, Poland

**Keywords:** antimalarial drugs, malaria, self-treatment, Ghana

## Abstract

*Introduction*: Malaria is still a huge social and economic health problem in the world. It especially affects the developing countries of Africa. A particular problem is the misuse and abuse of over-the-counter antimalarials. This problem could lead to the emergence of drug-resistant strains and the subsequent elimination of more antimalarials from the list of effective antimalarials in Ghana. *Methods*: During the implementation of the study, an original questionnaire was used to collect data among Ghanaians on their knowledge of malaria, attitude towards antimalarials and their use of antimalarials. *Results*: The proportion in the analyzed subgroups was compared using the chi-square test. The analysis was conducted using TIBCO Software Inc., Krakow, Poland (2017) and Statistica (data analysis software system), version 13. In total, 86.29% of respondents knew the symptoms of malaria (*p* = 0.02) and 57.2% knew the cause of malaria (*p* < 0.001). Respondents with higher education were significantly more likely to know the symptoms of malaria (96%) *p* < 0.001. In the study group, only 74.59% of the respondents consulted medical personnel before taking the antimalarial drug (*p* = 0.51) and only 14.2% of the remaining respondents performed a rapid diagnostic test for malaria. *Conclusions*: The awareness of Accra and Yendi native inhabitants about the causes and symptoms of malaria and alternative ways of prevention is quite high. People’s education very significantly influences the way Accra residents deal with suspected malaria. Widespread public education and awareness and accessibility to places where antimalarial drugs are sold play a very important role in the proper use of antimalarial drugs.

## 1. Introduction

Malaria is one of the commonest (apart from coronavirus disease-2019 (COVID-19), human immunodeficiency virus/acquired immunodeficiency syndrome (HIV/AIDS), and tuberculosis) infectious diseases in the world. According to the World Health Organization (WHO), around half of the world’s population lives in the endemic areas. In 2021, in 84 malaria endemic countries, there were an estimated 247 million malaria cases with increasing tendency. More than 5 million cases were reported in Ghana. The WHO reports 619,000 cases of malaria caused deaths globally with around 593,000 deaths in the African region [1].

Ghana is one of the most populous developing West African countries located in the *Plasmodium*-endemic areas with 28,206,728 inhabitants, and has the highest gross domestic product (GDP) in US dollars (USD) [2]. It is a country with a high incidence of malaria (266.4/1000 inhabitants in 2015) and high mortality among children under 5 (60/1000 live births in 2014). According to data from the Centers for Disease Control (CDC), Atlanta, Georgia, USA, malaria is the third-highest cause of death in Ghana after respiratory diseases and strokes, and is responsible for 8% of general mortality [3]. The data published by the WHO show that the number of new cases increases year by year [1].

Chemoprophylaxis among the inhabitants of Ghana is rarely used due to the serious side effects of long-term use. This is not the only reason. The WHO does not recommend chemoprophylaxis in African residents but recommends alternative strategies: intermittent preventive treatment in pregnant women (also in infants; IPTp and IPTi) and seasonal malaria chemoprevention (SMC) in young children living in the Sahel. Since 2021, the WHO has recommended the RTS, S/AS01 (Mosquirix^®^) malaria vaccine manufactured by GlaxoSmithKline (GSK) to prevent P. falciparum malaria in children living in areas with moderate-to-high malaria transmission [1].

The long-term use of monotherapy during malaria treatment has resulted in drug resistance. Since 2001, the WHO has recommended the use of artemisinin-based combination therapy (ACT) in countries facing the resistance problem [4]. Since 2002, ACT has been used in Ghana. Artesunate–amodiaquine became the first line of treatment for uncomplicated malaria. In cases where the use of this drug is impossible, it is recommended to use artemether–lumefantrine or dihydroartemisinin/piperaquine [5]. These drugs are available in pharmacies, where they are distributed by qualified personnel, but also in drug stores, on bazaars, or from hawkers, where the sale is dealt with by people without formal training [6]. The prices of all antimalarial drugs do not exceed USD 2.

Ghana was the first country to receive funding from the Global Fund for ACT. ACT prices stabilised at USD 0.5–USD 0.75. The price reduction increased the availability of ACT from 31% to 83% [7]. Information campaigns organised by the Affordable Medicines for Malaria Facility (AMFm) increased the use of ACT by physicians. The implementation of combined treatment with artemisinin and co-financing for medicines resulted in a significant decrease in mortality from malaria [8]. According to reports by the WHO and United Nations International Children’s Emergency Fund (UNICEF), it was estimated that, in 2000–2015, the number of deaths due to malaria decreased by 60%. Excessive use and self-administration of antimalarial drugs (or ACTs) lead to a risk of resistance to artemisinin and its partner drugs without medical consultations, and may expose the patients to adverse effects of ACTs. It has been reported that artemisinin-resistant *Plasmodium falciparum* strains have emerged and spread in Southeast Asia [9].

According to WHO guidelines [10], the main principles of malaria treatment are: early diagnosis and effective treatment, the rational use of antimalarial agents, the use of combined therapy, and the use of doses based on the weight of the patient.

In Ghana, the main diagnostic criterion is the presence of fever [11] and the majority of malaria diagnoses in these regions are based only on this symptom [6], which is nonspecific and characterises hundreds of different diseases. In 2012, the WHO launched the promotion of the T3 campaign (Test, Treat, Track). Each confirmed case should be treated with a quality-proven antimalarial drug and the disease should be monitored with appropriate surveillance systems [12].

Antimalarial drugs should only be used in confirmed cases of malaria. Wide access to an initial diagnosis of malaria was made possible by the rapid diagnostic test (RDT) [13]. The tests are quick and easy to use and are available at almost every pharmacy and in drug stores.

The aim of the study was to check the level of knowledge about the causes, symptoms, and alternative means of preventing malaria in Ghana in the cities of Accra and Yendi.

## 2. Materials and Methods

A survey was conducted in Accra, the capital of Ghana, and the city of Yendi located in northeastern Ghana. As a research tool, a specially prepared paper questionnaire was used, which was distributed in churches located in these cities; the inclusion criteria were: age above 18 years old, male and female, and ability to read and understand the English language. The questions in the questionnaire were mainly concerning the knowledge about malaria and antimalarial prophylaxis, antimalarial drugs, and attitudes to antimalarial drugs.

The survey was carried out in February and March 2017. Before the beginning of the research, the consent of the Bioethical Committee of the Nicolaus Copernicus University in Torun at the Collegium Medicum in Bydgoszcz (KB 742/2016 of 13/12/2016) was obtained. The questionnaire was anonymous and its completion was voluntary. The study group consisted of 248 people.

### Statistical Analysis

The data were analyzed using descriptive statistics. The proportion in the analyzed subgroups was compared using the chi-square test. A *p*-value < 0.05 was considered to be statistically significant. The analysis was conducted using TIBCO Software Inc., Krakow, Poland (2017). Statistica (data analysis software system), version 13 (http://statistica.io, accessed on 15 September 2022) was also used.

## 3. Results

Table 1 shows:The distribution of respondents by sex and age groups.Contact with malaria among family members.Knowledge of malaria symptoms.Knowledge of the cause of malaria by age.

**Table 1 medicina-59-00257-t001:** Knowledge about malaria depending on the age of the respondents.

Age Group	Distribution of People by Sex in Age Groups		Contact with Malaria among Family Members		Knowledge of Malaria Symptoms		Knowledge of the Cause of Malaria by Age	
	Male	Female	Yes	No	Yes	No	Yes	No
<16	12	28	37	3	31	9	11	29
16–20	18	31	45	4	44	5	31	18
21–30	39	49	81	7	82	6	49	39
31–40	28	17	42	3	39	6	31	14
41–50	8	11	14	5	13	6	15	4
>51	3	4	6	1	5	2	5	2

The entire study group consisted of 248 people, including 140 women (564,556.5%) and 108 men (435,543.6%).

A total of 90.7% of the respondents had contact with malaria among family members. There was no statistically significant relationship with the age of the respondents (*p* = 0.18).

A total of 86.3% of the respondents knew the symptoms of malaria. There was no significant statistical relationship between knowledge of malaria symptoms and age (*p* = 0.02).

Young people under 16 years of age were the least educated about the causes of malaria, and those over 40 years of age had the most knowledge (*p* < 0.001).

More than half of the respondents had completed secondary school. There were no significant statistical differences in the level of education in relation to gender (*p* = 0.84) (Figure 1).

A total of 89.1% of the respondents lived within 5 km of a store where they could purchase an antimalarial drug.

A total of 74.59% of the subjects consulted with the medical staff before taking the antimalarial drug and 25.41% did not consult. There was no relationship between the frequency of consultation and age (*p* = 0.51) (Figure 2).

Those with higher education were significantly more likely to have consulted with medical staff before taking an antimalarial drug (*p* < 0.001) (Figure 3).

A total of 57.2% of respondents knew the cause of malaria; 55.2% of respondents who knew the cause of malaria had completed at least primary school (Figure 4).

Respondents with higher education were significantly more likely to know the symptoms of malaria (96%) *p* < 0.001.

More than 20% of the respondents used the drug because of fever and headache; 4.4% of the respondents took the antimalarial drug only because they were afraid of malaria and had no symptoms of concern. There was no significant correlation between the reason for use and age (*p* = 0.93) (Figure 5).

Statistically significant differences were found in the age groups 21–30 years old and 31–40 years old with a significance level of *p* < 0.001. Among the respondents who did not know any alternative method of malaria prevention, statistically significant differences were found for the age group below 16 years old (*p* < 0.001) (Figure 6).

A total of 88.63% of the respondents who were not aware of alternative methods of malaria prevention completed at most primary school. Statistically significant differences were found among groups with different education level (*p* < 0.001) (Figure 7).

The respondents who did not consult their doctor before taking an antimalarial drug were asked if they were aware about the possibility of purchasing/performing a rapid diagnostic test for malaria.

Table 2 shows:Knowledge of how to purchase and perform a rapid diagnostic test for malaria (before using the drug without consulting medical staff).Performing a rapid diagnostic test for malaria before purchasing antimalarial drugs.

**Table 2 medicina-59-00257-t002:** Knowledge about malaria tests by age of respondents.

Age Group	Knows about the Possibility to a Purchase Rapid Test for Malaria		Did the Respondents Take a Rapid Test for Malaria before Purchasing Antimalarial Drugs?	
	Yes	No	Yes	No
<16	2	7	1	15
16–20	6	5	2	9
21–30	11	6	2	9
31–40	9	7	3	12
41–50	5	3	1	7
>51	1	1	0	2

More than half of the respondents who used the drug without consulting medical personnel knew that they could purchase and take a rapid diagnostic test for malaria. There was no significant statistical relationship with the age of the respondents (*p* = 0.45).

Only 14.2% of the respondents who did not consult the inclusion of antimalarial treatment with medical staff took a rapid diagnostic test for malaria. There were no significant statistical differences in the study group by age, *p* = 0.87.

For 46.03% of the respondents in the group who did not consult medical personnel about taking the antimalarial drug, the test was too expensive; for 44.4%, a family suggestion dissuaded them from purchasing the test. For 17.46%, a strong fear of malaria deterred them from purchasing the test. Age had no effect on the reason for not taking a rapid diagnostic test for malaria, *p* = 0.26 (Figure 8).

The average monthly income in Ghana for 77.8% of respondents did not exceed USD 20. 10.9% earned USD 20.1–50; 13.7% earned USD 50.1–100; and only 7.4% earned more than USD 100.

## 4. Discussion

### 4.1. Knowledge of Malaria Symptoms

In the study we conducted in Accra and Yendi, 86.29% of the respondents knew the symptoms of malaria. In an earlier study conducted by Amposah et al. in Ghana, only 57.3% of respondents indicated headache, body aches, fever, and loss of appetite as malaria symptoms [14]. The results of the study conducted by Buabeng indicated that only 24% of the participants knew the symptoms typical of malaria, indicating the need to report to a health facility [15]. In contrast, data collected among residents of Bolgatanga (Municipal District and Upper East Region of Ghana) who had undergone an episode of malaria indicated that 96.9% knew the symptoms of malaria [16]. Our respondents most frequently cited fever and headache (69.75%). Additionally, of concern is the fact that almost 8% of respondents reached for antimalarial drugs because of fear of malaria, even when they had no worrisome symptoms.

### 4.2. Knowledge of the Cause of Malaria

In the study, 57.2% of the respondents knew the cause of malaria. Among that group, 96.5% had completed at least primary school. In a study conducted by Aborah and colleagues, 75% of the respondents knew the causes of malaria and 93.1% knew the mode of transmission of malaria [16]. Amposah’s study indicated that 98.7% knew the modes of malaria transmission and 68.3% of the respondents knew the three methods of malaria prevention [14].

Kwame Buabeng’s results showed no significant difference in knowledge of the cause of malaria in relation to the education level [15]. In our study, age and education played a significant role in having knowledge about the cause of malaria. Those with secondary and high education had good knowledge about the cause of malaria compared to the other groups. A total of 86.6% of those under 16 years of age had the least knowledge about the cause of malaria (*p* < 0.001). In the study by Abate, most of the respondents (85.2%) attributed the cause of malaria to a mosquito bite. However, in the study conducted in the Ho municipality, 40% of the respondents answered that malaria was transmitted from person to person [17]. Moreover, in Ayi’s study, more than 20% of the respondents attributed the cause of malaria to chewing maize stalk, starvation, lack of hygiene, and exposure to cold. People who had not been educated about malaria believed that malaria could be caused, for example, by eating mangoes, by sunshine, or by drinking dirty water [18].

### 4.3. Knowledge of Malaria Prevention

A study by Amposah indicated that 68.3% of respondents knew three methods of malaria prevention [14]. In the study by Singh, 90% of the respondents had the knowledge that mosquito nets were effective in preventing malaria, but only 80% used them. In a study conducted by Konlan in the Ho district, about 54% and 20% of the respondents had satisfactory and good levels of knowledge about malaria prevention, respectively [17].

According to Azabre in Kassena-Nankana, 62% of respondents had insecticide-treated nets, but only 27% used them in windows and doors [19]. Despite the awareness that fresh water was a mosquito reservoir, about 50% of the households had open-water reservoirs and more than 70% had open-water tanks [19]. A total of 72.2% of the respondents in the group who knew at least one malaria prevention method had a minimum of secondary education and those who knew two prevention methods were dominated by those with higher education. This is another example of the lack of public awareness and effective use of the available tools for malaria prevention. As Konlan points out, the main sources of information about malaria among the Ghanaian population are television (74%), health workers (66%), schools (62%), family and friends (60%), and the internet (51%) [17].

### 4.4. Consultation with Medical Personnel before Taking Antimalarial Drugs

The strategy of malaria treatment recommended in the WHO guidelines [10] seems logical, but Ghana’s health conditions, average monthly earnings, and public awareness are a little different than we think. Public health insurance (NHIS) is available in Ghana. The basic package gives free access to health care to treat the most common illnesses in Ghana, including malaria. The insurance covers the costs of doctor’s visits, hospitalisation, and medical expenses related to this, but does not cover the cost of purchasing medicines. Due to the relatively high costs of purchasing insurance in relation to remuneration and low access to health care facilities in rural areas, mainly the communities in big cities buy National Health Insurance System (NHIS) insurance policies. Even in the most prosperous communities, more than a third of the older population is not covered by NHIS [20]. The remaining part of the society may use medical care only against full payment. The cost of a medical visit is on average around 5 USD, which is approx. 20–25% of the monthly salary of the poorest.

The cost of treatment is not the only problem. Ghana has serious deficits in medical staff. According to WHO data, in 2017, the number of doctors was only 0.136/1000 inhabitants [2]. Most doctors work in large and medium-sized cities. In small health centres located in rural areas, there are only nurses and physician assistants [21]. Ghana belongs to the countries with the lowest number of nurses; in 2018, there were only 4.2 nurses/1000 inhabitants [2], which means that there are areas where even basic health services are unavailable. In 2013, over 2400 pharmacies and over 10,000 licensed chemical stores were registered in Ghana [22].

In the study group, the average monthly income of 67.96% was USD 20 which effectively discourages people from seeking medical consultation. According to Amponsah, 28.2% of the respondents used prayer instead of taking medication. According to the same author, 50.3% of the respondents believed that doctors depend too much on medicines which also discourages them from taking prescribed medicines [14]. In the study group, only 74.59% of the respondents reported that they consulted their treatment with medical staff. The high cost of medical visits, the remoteness of medical facilities, and the lack of trust in doctors in the community are some of the reasons for taking antimalarial drugs without medical consultation [15].

In the study group, 65.72% of the respondents with a minimum of primary school education consulted with medical personnel to take antimalarial drugs. It is very clear to what extent education influences patients’ awareness before determining the appropriate management. In the group of respondents with higher education, 96.22% consulted medical personnel before taking antimalarial drugs. The well-developed network of stores where antimalarial drugs can be bought makes it easier to buy them without a prescription. In the study group, 89.11% of the respondents lived within 5 km of a store where they could buy the drug. Few vendors process the prescription [6].

In addition, some authors report that staff at chemical stores are more shopper-friendly than staff at pharmacies, which also favours shopping at stores located where people live [23].

### 4.5. Rapid Diagnostic Tests for Malaria

Rapid diagnostic tests are available in almost every pharmacy. The cost of a diagnostic test is on average 2–3 times higher than the price of an antimalarial drug, which is one of the factors limiting their use in Ghana. In the study group, among those who had not consulted medical personnel for treatment, 53.96% knew about the possibility of purchasing a rapid diagnostic test for malaria, but only 14.28% of the respondents had done so. According to our survey, 46% of the respondents thought the tests were too expensive. About 15% of the respondents took the drug without testing because of fear of malaria. A similar group took the drug without confirming the test because they had the drug at home. In Aborah’s study in Bolgatanga, 43.4% of the respondents used nonprescription drugs before visiting a health facility (16).

### 4.6. Appropriate Use of Antimalarial Drugs

The use of antimalarial drugs by people who purchased them from pharmacies and clinics is more rational than the use by people who purchased drugs from chemical stores and people who treated themselves with leftover drugs they had at home [15]. Such inappropriate use may be due to the inadequate training of people who market antimalarial drugs [15].

Kwame Buabeng’s study indicated that there was a high prevalence of antimalarial drug use in the community before going to hospital, and that in most cases, these drugs were used inappropriately [15].

During a study conducted at a community pharmacy in Lakeside Estate, it was estimated that only 57.33% of the respondents were taking the drugs according to the recommended dosage. Subtherapeutic doses may lead to drug resistance [14]. Despite the availability of “Western” medicines, some patients preferred topical herbal remedies [24]. They mainly purchased medicines from local vendors, choosing products that were most heavily advertised [24].

## 5. Conclusions

(1)Awareness among the people of Accra and Yendi about the causes, symptoms, and prevention of malaria is quite high.(2)Public education and awareness, as well as accessibility to places where antimalarial drugs are sold, play a very important role in the proper use of antimalarial drugs.

## Figures and Tables

**Figure 1 medicina-59-00257-f001:**
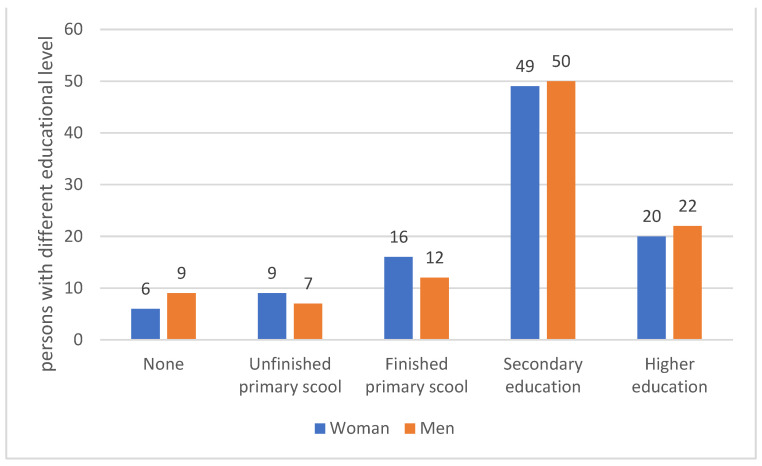
Educational level of the respondents.

**Figure 2 medicina-59-00257-f002:**
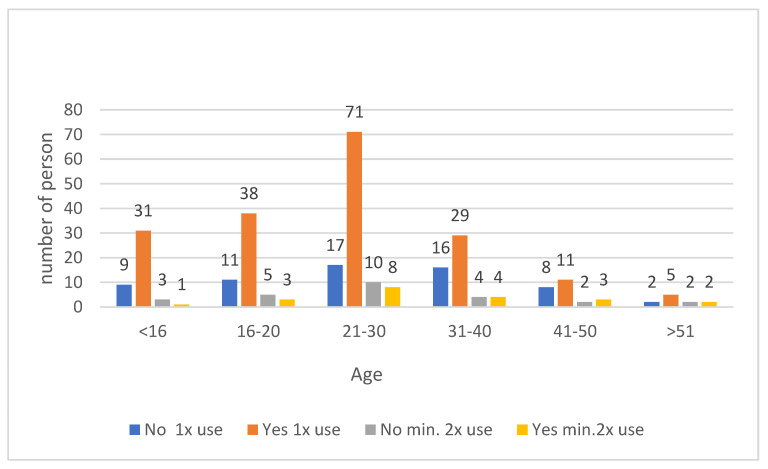
Use of antimalarial drugs without consultation with medical personnel across age groups.

**Figure 3 medicina-59-00257-f003:**
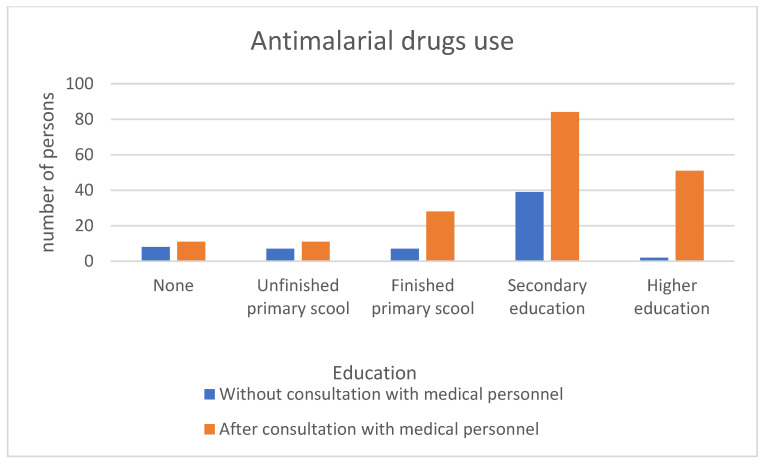
Use of antimalarial drugs without/after consultation with medical personnel according to education level.

**Figure 4 medicina-59-00257-f004:**
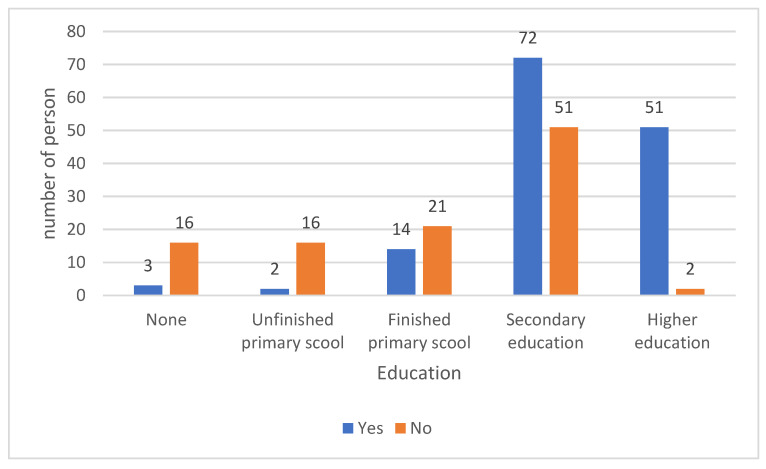
Number of people knowing the cause of malaria by education.

**Figure 5 medicina-59-00257-f005:**
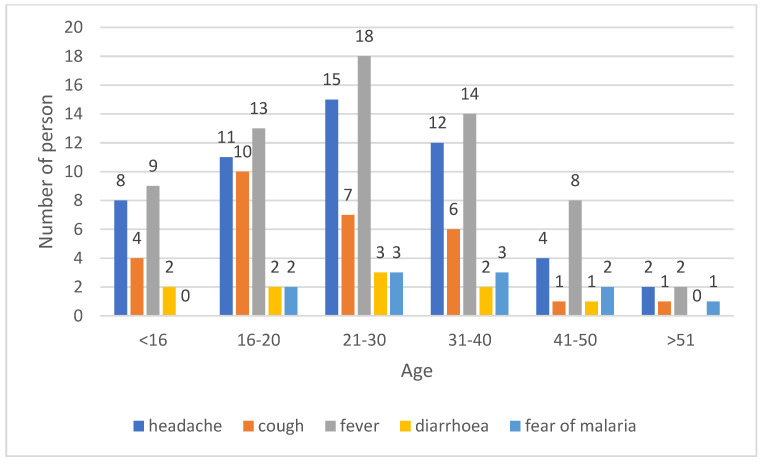
Respondents answers to the question: why did you use an antimalarial drug?

**Figure 6 medicina-59-00257-f006:**
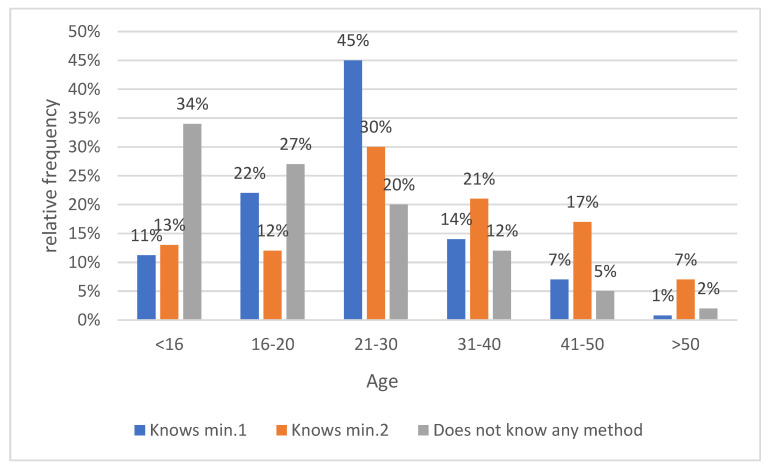
Knowledge of alternative methods to prevent malaria.

**Figure 7 medicina-59-00257-f007:**
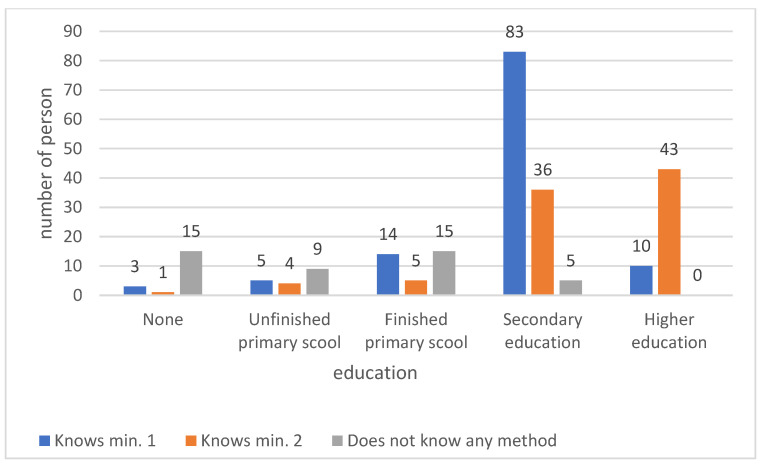
Knowledge of alternative methods of malaria prevention according to education.

**Figure 8 medicina-59-00257-f008:**
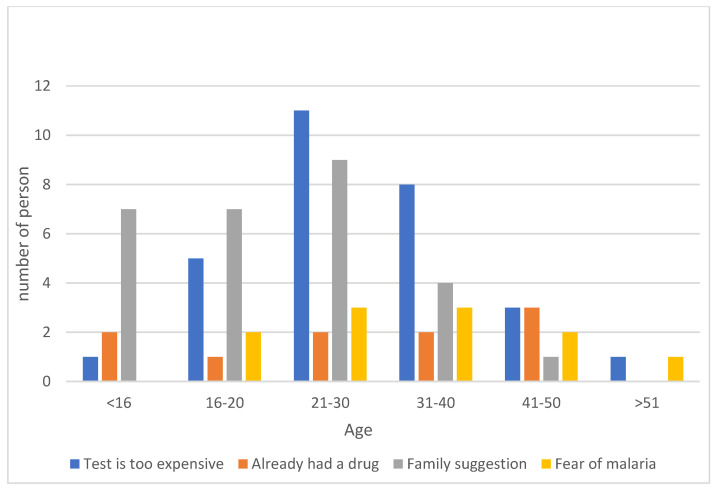
Results of responses to the question of why the respondent did not take a rapid diagnostic test for malaria.

## Data Availability

All data and materials are available upon request from the editorial office.

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
