# Peer review of "The Problem of Antimalarial-Drug Abuse by the Inhabitants of Ghana"

_medicina, 2023, doi:10.3390/medicina59020257_

Round 1

Reviewer 1 Report

It is an interesting and important article as it concerns a real problem in a continent where malaria is a real public health problem and correct use of medicines is not always practiced, thus increasing the appearance of resistance from the parasite.

Here are some comments:

1. Pag 1, lines 39-41: please, replace with more recent data also in other parts of the text. The World Malaria Report 2022 has been published.

2. Pag 1, line 43: please, write Plasmodium in Italics.

3. Pag 2, line 54: please, capitalize the initials of "Combination Therapy".

4. Pag 2, line 83: please, capitalize the initials of "Rapid Diagnostic Tests” and add “diagnostic” in parenthesis.

5. Pag 2, line 90: point missing.

6. Pag 8, line 157: point missing.

7. Pag 9, line 176: point missing.

8. Pag 10, line 187: point missing.

9. Figure 2: please, write “percent of educated people”.

10. Figure 11: please, better explain what is written in the legend (Knows min.1..... etc.).

11. Figure 12: please, substitute “numbero” with “number”.

12. Figure 13- page 9, line 168: please, close the parenthesis before the period.

13. Pag 9, line 176: point missing.

14. Pag 10, line 187: point missing.

15. Pag 11, line 235: please, put a space between "wetlands" and (19).

16. Pag 12, line 256: please, move the dot after the bibliography.

17. Pag 12, line 262: please, put a space between "assistants" and (26).

18. Pag 12, line 267: please, put a space between "200" and “GHC”.

19. Pag 13, line 304: please, put a space between "drugs" and “(17)”.

20. Pag 13: reference number 30 is not mentioned. Please check.

Author Response

Dear Reviewer,

Thank you very much for your accurate comments in the presented review.

All suggested changes have been made in our article.

1. Pag 1, lines 39-41: please, replace with more recent data also in other parts of the text. The World Malaria Report 2022 has been published. – The data has been updated

2. Pag 1, line 43: please, write Plasmodium in Italics. - has been changed

3. Pag 2, line 54: please, capitalize the initials of "Combination Therapy". - has been changed

4. Pag 2, line 83: please, capitalize the initials of "Rapid Diagnostic Tests” and add “diagnostic” in parenthesis. - has been changed

5. Pag 2, line 90: point missing. - has been changed

6. Pag 8, line 157: point missing. - has been changed

7. Pag 9, line 176: point missing. - has been changed

8. Pag 10, line 187: point missing. - has been changed

9. Figure 2: please, write “percent of educated people”. - has been changed

10. Figure 11: please, better explain what is written in the legend (Knows min.1..... etc.). - explanation: knows at least one method, knows at least two methods

11. Figure 12: please, substitute “numbero” with “number”. - has been changed

12. Figure 13- page 9, line 168: please, close the parenthesis before the period. - has been changed

13. Pag 9, line 176: point missing. - has been changed

14. Pag 10, line 187: point missing. - has been changed

15. Pag 11, line 235: please, put a space between "wetlands" and (19). - has been changed

16. Pag 12, line 256: please, move the dot after the bibliography. - has been changed

17. Pag 12, line 262: please, put a space between "assistants" and (26). - has been changed

18. Pag 12, line 267: please, put a space between "200" and “GHC”. - has been changed

19. Pag 13, line 304: please, put a space between "drugs" and “(17)”. - has been changed

20. Pag 13: reference number 30 is not mentioned. Please check. - reference has been removed - there is no reference to this reference in the text

Reviewer 2 Report

The authors conducted a KAP survey in 248 persons residing in two cities in Ghana in 2017. The introduction is adequate. The methods section needs more information. The presentation of results (figures) can be improved (see my major comments). Discussion should be focused on works done in Ghana.

Please explain what the abbreviations stand for when first used in the text.

Complete references should be given. English needs many corrections.

MAJOR COMMENTS:

Lines 91-104, Methods: The authors should provide more details. What were the survey questions about? How were the respondents chosen (age, sex, etc.)? Was there an interview or was it a written questionnaire that the respondents answered? Although some of the answers to these questions are given in the Results section, more information about the methods should be provided. The authors may also consider providing a copy of their questionnaire as a Supplementary Material.

Results, general and specific comments on the figures: There are too many figures (16 in all). Some figures do not contain a lot of data. Data presented in several figures can be grouped together into a single table. For example, figures with age groups and numbers of persons (Fig. 1, 3, 4, 8) and others (without age groups) can be grouped into a single table.

Many of the minor comments concerning Fig. 1 to 3 (see below) also apply to other figures. Moreover, data in some figures can be described in the text. For example, for Fig. 5, lines 124-125 describe very well the results in the figure. Fig. 5 can be deleted.

Fig. 1: Y-axis legend – number of persons; Figure title – I suggest, distribution of respondents by sex and age groups; The figure title “Distribution of people by sex in age groups” embedded in the figure can be deleted.

Fig. 2: title – I suggest – educational level of the respondents; Y-axis legend, I suggest – persons with different educational level (%); The term “secondary education” is not clear. Does it mean that these persons graduated from high school? The term “education” embedded in the figure can be deleted.

Fig. 3: The figure title embedded in the figure (“Did you have a contact with malaria at your family?”) can be deleted. Y-axis legend – number of persons; The expression “Contact with malaria” is not clear to me. Does it mean the presence of at least one family member with confirmed malaria at the time of survey? Please explain.

Fig. 6: It is not clear how the question was asked about antimalarial drug use. Does self-medication refer to the past few months or years?

Fig. 16: earnings. This figure does not contain any data that have relations with other factors that were analyzed. It can be deleted and the essential information given in the text.

Discussion: Comparison of the results of the present study with those of other studies performed earlier in Ghana is helpful. However, the authors also compare their results with those of other authors in other African countries. Unless the comparison with other countries is justified, I think that the authors should focus on works done in Ghana. 

MINOR COMMENTS:

Line 37: coronavirus disease-2019 (COVID-19), human immunodeficiency virus/acquired immunodeficiency syndrome (HIV/AIDS)

Line 38: World Health Organization (WHO)

Line 41, REF 1: Please cite the latest available WHO World Malaria Report 2022. (https://www.who.int/teams/global-malaria-programme/reports/world-malaria-report-2022)

Lines 42-43: 28,206,728 inhabitants; US dollars (USD); gross domestic product (GDP); Plasmodium (in italic)

Line 45, “high incidence of malaria and high mortality among children under 5”: Please add a reference citation to support these data (unless they are in REF 2).

Line 46: Centers for Disease Control (CDC), Atlanta, Georgia, USA

Line 46: Please re-check that stroke is a major killer in Ghana. How about gastrointestinal diseases (diarrhea)? REF 3 refers to CDC. Any data published by the Ghanaian Ministry of Health?

Line 49, “chemoprophylaxis among the inhabitants of Ghana is rarely used due to serious side effects of long-term use”: This is not the only reason. WHO does not recommend chemoprophylaxis in African residents in Africa but recommends alternative strategies: intermittent preventive treatment in pregnant women (also in infants; IPTp and IPTi) and seasonal malaria chemoprevention (SMC) in young children living in the Sahel.

Line 52: resulted in drug resistance

Lines 53-54: Since 2001, the WHO recommends the use of artemisinin-based combination therapy (ACT)…

Lines 55-56: Artesunate-amodiaquine became the first-line treatment for uncomplicated malaria.

Line 59, line 284, and elsewhere, “chemical stores”: Is this the way Ghanaian people call these stores? Isn’t it “drug stores”?

Line 62, “ACT treatment”: ACT stands for artemisinin-based combination therapy. Saying “artemisinin-based combination therapy treatment” is redundant. “Treatment” can be deleted.

Line 64: Affordable Medicines for Malaria Facility (AMFm)

Line 67: United Nations International Children’s Emergency Fund (UNICEF)

Lines 68-69: Excessive use and taking drugs (verb in plural form); for example, excessive use and self-administration of antimalarial drugs (or ACTs) lead to a risk of resistance to artemisinin and its partner drugs; delete “may be the cause of complications to antimalarial medications” to change to may expose the patients to adverse effects of ACTs

Line 71-72, “reports by other researchers suggest that…”: Artemisinin resistance has been documented (not only “suggested”). I suggest - It has been reported that artemisinin-resistant Plasmodium falciparum strains have emerged and spread in southeast Asia

Line 76: criterion is fever OR criterion is the presence of fever

Line 83, “…the launch of RDTs (rapid tests that diagnose malaria)”: revise to – made possible by rapid diagnostic test (RDT)

Lines 85-86, “The tests are quick and easy to use…”: This sentence alone should not be a paragraph. It can be placed in line 84 as part of the preceding paragraph.

Lines 87-90: delete line 87 “the aim of the study”; delete “1)” because there is no “2”; The sentence “The aim of the study was to evaluate the level of knowledge about the causes…” can be placed at the end of the previous sentence as part of the paragraph that starts with “Antimalarial drugs should only be used…”

Lines 106-107: The percentages can be expressed to the first decimal place: 56.5% and 43.6%. This remark applies to the entire text.

Line 120: 86.3%

Lines 138-139, Fig. 8: What is the “knowledge of the cause of malaria”? What was the question asked? Were there pre-determined multiple choice answers?

Line 140, Fig. 9: Please delete the Polish word in the figure.

Lines 154-155, “statistically significant differences were found…”: Please add what the authors are talking about in this sentence (knowledge of alternative methods).

Lines 191-193, “Lower results were observed…Erhun”: This was a study conducted in Nigeria. The authors should focus on Ghana. This sentence (and the reference citation) can be deleted.

Line 193: In an earlier study conducted by Amposah et al. in Ghana…

Lines 194-196: The results of another study conducted in Ghana by Buabeng et al. indicate (not “indicates”) that…

Line 197: where is “Bolgatanga”? Please indicate in the text that the studies being cited were conducted in Ghana. The region or city where the studies were conducted (REF 16, 17, 18) can also be specified in the text.

Lines 198-199, lines 202-203, “in the Abate study in Ethiopia…”: Please focus on Ghana. Studies performed elsewhere do not need to be cited unless really necessary for data interpretation of the present work.

Lines 203-204, “in a study conducted by Singh”: Again, this is a study performed in Nigeria.

Line 213, “age and education plays”: age and education play

Line 247, “the tactic malaria treatment”: The strategy of malaria treatment recommended in the WHO guidelines

Line 249: National Health Insurance System (NHIS)???

Line 260: 0.136

Line 263: lowest number of nurses; 4.2 nurses

Line 267: income was

Line 277 and elsewhere: what is “p/malarial drugs”?

Line 277: to what extent

Lines 297-298, “according to Amponsah… (29)”: Amponsah et al. is REF 16. Please check. REF 29 was a study conducted in the United States. I think that this reference citation is irrelevant to the present study in Ghana.

Line 299, “only 32.5% of the population buys herbal antimalarial drugs…(32)”: REF 32 is a study performed in Haiti. It is not relevant to the situation in Ghana. Please cite a more appropriate reference.

Line 317: prevention of malaria

REF 4 is incomplete: Ashley EA, Pyae Phyo A, Woodrow CJ. Malaria. Lancet. 2018;391(10130):1608-1621. doi: 10.1016/S0140-6736(18)30324-6.

REF 6: Please provide more information (web link, for example).

REF 7 is incomplete (page numbers): Goodman C, Brieger W, Unwin A, Mills A, Meek S, Greer G. Medicine sellers and malaria treatment in sub-Saharan Africa: what do they do and how can their practice be improved? Am J Trop Med Hyg. 2007;77(6 Suppl):203-218.

REF 8 is incomplete (page numbers): Malm KL, Segbaya S, Forson I, Gakpey KD, Sampong LB, Adjei EA, Bart-Plange C. Implementation of the Affordable Medicines for Malaria Facility (AMFm) in Ghana: processes, challenges and achievements. J Public Health Policy. 2013;34(2):302-314. doi: 10.1057/jphp.2013.12.

REF 9 is incomplete (page numbers): Ndeffo Mbah ML, Parikh S, Galvani AP. Comparing the impact of artemisinin-based combination therapies on malaria transmission in sub-Saharan Africa. Am J Trop Med Hyg. 2015;92(3):555-60. doi: 10.4269/ajtmh.14-0490.

REF 10: Tun KM, Imwong M, Lwin KM, Win AA, Hlaing TM, Hlaing T, Lin K, Kyaw MP, Plewes K, Faiz MA, Dhorda M, Cheah PY, Pukrittayakamee S, Ashley EA, Anderson TJ, Nair S, McDew-White M, Flegg JA, Grist EP, Guerin P, Maude RJ, Smithuis F, Dondorp AM, Day NP, Nosten F, White NJ, Woodrow CJ. Spread of artemisinin-resistant Plasmodium falciparum in Myanmar: a cross-sectional survey of the K13 molecular marker. Lancet Infect Dis. 2015;15(4):415-421. doi: 10.1016/S1473-3099(15)70032-0.

REF 11 seems to be wrong cited. This WHO document was not published by Trans R Soc Trop Med Hyg. World Health Organization. (‎2015)‎. Guidelines for the treatment of malaria, 3rd ed. Geneva, World Health Organization. https://apps.who.int/iris/handle/10665/162441 . Moreover, a more recent WHO document is available. Please see the web link: https://reliefweb.int/report/world/who-guidelines-malaria-3-june-2022

REF 13 is available on line. Please add the web link.

REF 14 is available on line: World Health Organization. (‎2012)‎. Test, treat, track: scaling up diagnostic testing, treatment and surveillance for malaria. World Health Organization. https://apps.who.int/iris/handle/10665/337979

REF 15: Please complete this reference: page numbers or article number.

REF 16: Amponsah AO, Vosper H, Marfo AF. Patient related factors affecting adherence to antimalarial medication in an urban estate in Ghana. Malar Res Treat. 2015;2015:452539. doi: 10.1155/2015/452539.

REF 17: Buabeng KO, Duwiejua M, Dodoo AN, Matowe LK, Enlund H. Self-reported use of anti-malarial drugs and health facility management of malaria in Ghana. Malar J. 2007;6:85. doi: 10.1186/1475-2875-6-85.

REF 18: Aborah S, Akweongo P, Adjuik M, Atinga RA, Welaga P, Adongo PB. The use of non-prescribed anti-malarial drugs for the treatment of malaria in the Bolgatanga municipality, northern Ghana. Malar J. 2013;12:266. doi: 10.1186/1475-2875-12-266.

REF 19 and 20: x Ethiopia/Nigeria

REF 22: Ayi I, Nonaka D, Adjovu JK, Hanafusa S, Jimba M, Bosompem KM, Mizoue T, Takeuchi T, Boakye DA, Kobayashi J. School-based participatory health education for malaria control in Ghana: engaging children as health messengers. Malar J. 2010;9:98. doi: 10.1186/1475-2875-9-98.

REF 23: Please add the authors and page numbers.

REF 24: x Ethiopia

REF 25: Van der Wielen N, Channon AA, Falkingham J. Universal health coverage in the context of population ageing: What determines health insurance enrolment in rural Ghana? BMC Public Health. 2018;18(1):657. doi: 10.1186/s12889-018-5534-2.

REF 28: Williams HA, Jones CO. A critical review of behavioral issues related to malaria control in sub-Saharan Africa: what contributions have social scientists made? Soc Sci Med. 2004;59(3):501-523. doi: 10.1016/j.socscimed.2003.11.010.

REF 29: Kemppainen J, Kim-Godwin YS, Reynolds NR, Spencer VS. Beliefs about HIV disease and medication adherence in persons living with HIV/AIDS in rural southeastern North Carolina. J Assoc Nurses AIDS Care. 2008;19(2):127-136. doi: 10.1016/j.jana.2007.08.006.

REF 30: Please check if REF 30 was cited in the main text.

REF 30: Bonful HA, Awua AK, Adjuik M, Tsekpetse D, Adanu RMK, Nortey PA, Ankomah A, Koram KA. Extent of inappropriate prescription of artemisinin and anti-malarial injections to febrile outpatients, a cross-sectional analytic survey in the Greater Accra region, Ghana. Malar J. 2019;18(1):331. doi: 10.1186/s12936-019-2967-8.

REF 31: Sci 2020;2(3):49. doi: 10.3390/sci2030049

REF 32: Druetz T, Andrinopoulos K, Boulos LM, Boulos M, Noland GS, Desir L, Lemoine JF, Eisele TP. "Wherever doctors cannot reach, the sunshine can": overcoming potential barriers to malaria elimination interventions in Haiti. Malar J. 2018;17(1):393. doi: 10.1186/s12936-018-2553-5.

Author Response

Dear Reviewer,

Thank You very much for constructive evaluation of our text. We also Thank You very much for helping us to make important corrections. As suggested by the Reviewer, we have made all the indicated corrections.

Please explain what the abbreviations stand for when first used in the text.

Complete references should be given. English needs many corrections.

MAJOR COMMENTS:

Lines 91-104, Methods: The authors should provide more details. What were the survey questions about? How were the respondents chosen (age, sex, etc.)? Was there an interview or was it a written questionnaire that the respondents answered? Although some of the answers to these questions are given in the Results section, more information about the methods should be provided. The authors may also consider providing a copy of their questionnaire as a Supplementary Material. - significant, missing information is described in the body of the article

Results, general and specific comments on the figures: There are too many figures (16 in all). Some figures do not contain a lot of data. Data presented in several figures can be grouped together into a single table. For example, figures with age groups and numbers of persons (Fig. 1, 3, 4, 8) and others (without age groups) can be grouped into a single table. - as suggested by the reviewer, the charts were changed into tables

Many of the minor comments concerning Fig. 1 to 3 (see below) also apply to other figures. Moreover, data in some figures can be described in the text. For example, for Fig. 5, lines 124-125 describe very well the results in the figure. Fig. 5 can be deleted. - We have made corrections as suggested by the reviewer

Fig. 1: Y-axis legend – number of persons; Figure title – I suggest, distribution of respondents by sex and age groups; The figure title “Distribution of people by sex in age groups” embedded in the figure can be deleted. - We have made corrections as suggested by the reviewer

Fig. 2: title – I suggest – educational level of the respondents; Y-axis legend, I suggest – persons with different educational level (%); The term “secondary education” is not clear. Does it mean that these persons graduated from high school? The term “education” embedded in the figure can be deleted. - We have made corrections as suggested by the reviewer

Fig. 3: The figure title embedded in the figure (“Did you have a contact with malaria at your family?”) can be deleted. Y-axis legend – number of persons; The expression “Contact with malaria” is not clear to me. Does it mean the presence of at least one family member with confirmed malaria at the time of survey? Please explain. - We have made corrections as suggested by the reviewer

Fig. 6: It is not clear how the question was asked about antimalarial drug use. Does self-medication refer to the past few months or years? - we have added important information

Fig. 16: earnings. This figure does not contain any data that have relations with other factors that were analyzed. It can be deleted and the essential information given in the text. - We have made corrections as suggested by the reviewer

Discussion: Comparison of the results of the present study with those of other studies performed earlier in Ghana is helpful. However, the authors also compare their results with those of other authors in other African countries. Unless the comparison with other countries is justified, I think that the authors should focus on works done in Ghana. - as suggested by the reviewer, we removed irrelevant references from the text

MINOR COMMENTS:

Line 37: coronavirus disease-2019 (COVID-19), human immunodeficiency virus/acquired immunodeficiency syndrome (HIV/AIDS) - we have made a correction

Line 38: World Health Organization (WHO) - we have made a correction

Line 41, REF 1: Please cite the latest available WHO World Malaria Report 2022. (https://www.who.int/teams/global-malaria-programme/reports/world-malaria-report-2022) - we have made a correction

Lines 42-43: 28,206,728 inhabitants; US dollars (USD); gross domestic product (GDP); Plasmodium (in italic) - we have made a correction

Line 45, “high incidence of malaria and high mortality among children under 5”: Please add a reference citation to support these data (unless they are in REF 2). - we have made a correction

Line 46: Centers for Disease Control (CDC), Atlanta, Georgia, USA - we have made a correction

Line 46: Please re-check that stroke is a major killer in Ghana. How about gastrointestinal diseases (diarrhea)? REF 3 refers to CDC. Any data published by the Ghanaian Ministry of Health? - we have made a correction

Line 49, “chemoprophylaxis among the inhabitants of Ghana is rarely used due to serious side effects of long-term use”: This is not the only reason. WHO does not recommend chemoprophylaxis in African residents in Africa but recommends alternative strategies: intermittent preventive treatment in pregnant women (also in infants; IPTp and IPTi) and seasonal malaria chemoprevention (SMC) in young children living in the Sahel. - we have made a correction

Line 52: resulted in drug resistance - we have made a correction

Lines 53-54: Since 2001, the WHO recommends the use of artemisinin-based combination therapy (ACT)… - we have made a correction

Lines 55-56: Artesunate-amodiaquine became the first-line treatment for uncomplicated malaria. - we have made a correction

Line 59, line 284, and elsewhere, “chemical stores”: Is this the way Ghanaian people call these stores? Isn’t it “drug stores”? - we have made a correction

Line 62, “ACT treatment”: ACT stands for artemisinin-based combination therapy. Saying “artemisinin-based combination therapy treatment” is redundant. “Treatment” can be deleted. - we have made a correction

Line 64: Affordable Medicines for Malaria Facility (AMFm) - we have made a correction

Line 67: United Nations International Children’s Emergency Fund (UNICEF) - we have made a correction

Lines 68-69: Excessive use and taking drugs (verb in plural form); for example, excessive use and self-administration of antimalarial drugs (or ACTs) lead to a risk of resistance to artemisinin and its partner drugs; delete “may be the cause of complications to antimalarial medications” to change to may expose the patients to adverse effects of ACTs - we have made a correction

Line 71-72, “reports by other researchers suggest that…”: Artemisinin resistance has been documented (not only “suggested”). I suggest - It has been reported that artemisinin-resistant Plasmodium falciparum strains have emerged and spread in southeast Asia - we have made a correction

Line 76: criterion is fever OR criterion is the presence of fever - we have made a correction

Line 83, “…the launch of RDTs (rapid tests that diagnose malaria)”: revise to – made possible by rapid diagnostic test (RDT) - we have made a correction

Lines 85-86, “The tests are quick and easy to use…”: This sentence alone should not be a paragraph. It can be placed in line 84 as part of the preceding paragraph. - we have made a correction

Lines 87-90: delete line 87 “the aim of the study”; delete “1)” because there is no “2”; The sentence “The aim of the study was to evaluate the level of knowledge about the causes…” can be placed at the end of the previous sentence as part of the paragraph that starts with “Antimalarial drugs should only be used…” - we have made a correction

Lines 106-107: The percentages can be expressed to the first decimal place: 56.5% and 43.6%. This remark applies to the entire text. - we have made a correction

Line 120: 86.3% - we have made a correction

Lines 138-139, Fig. 8: What is the “knowledge of the cause of malaria”? What was the question asked? Were there pre-determined multiple choice answers? - we have made a correction

Line 140, Fig. 9: Please delete the Polish word in the figure. - we have made a correction

Lines 154-155, “statistically significant differences were found…”: Please add what the authors are talking about in this sentence (knowledge of alternative methods). - we have made a correction

Lines 191-193, “Lower results were observed…Erhun”: This was a study conducted in Nigeria. The authors should focus on Ghana. This sentence (and the reference citation) can be deleted. - we have made a correction

Line 193: In an earlier study conducted by Amposah et al. in Ghana…- we have made a correction

Lines 194-196: The results of another study conducted in Ghana by Buabeng et al. indicate (not “indicates”) that…- we have made a correction

Line 197: where is “Bolgatanga”? Please indicate in the text that the studies being cited were conducted in Ghana. The region or city where the studies were conducted (REF 16, 17, 18) can also be specified in the text. - we have made a correction

Lines 198-199, lines 202-203, “in the Abate study in Ethiopia…”: Please focus on Ghana. Studies performed elsewhere do not need to be cited unless really necessary for data interpretation of the present work. - we have made a correction

Lines 203-204, “in a study conducted by Singh”: Again, this is a study performed in Nigeria. - we have made a correction

Line 213, “age and education plays”: age and education play - we have made a correction

Line 247, “the tactic malaria treatment”: The strategy of malaria treatment recommended in the WHO guidelines - we have made a correction

Line 249: National Health Insurance System (NHIS)??? - we have made a correction

Line 260: 0.136 - we have made a correction

Line 263: lowest number of nurses; 4.2 nurses - we have made a correction

Line 267: income was - we have made a correction

Line 277 and elsewhere: what is “p/malarial drugs”? - we have made a correction

Line 277: to what extent - we have made a correction

Lines 297-298, “according to Amponsah… (29)”: Amponsah et al. is REF 16. Please check. REF 29 was a study conducted in the United States. I think that this reference citation is irrelevant to the present study in Ghana. - we have made a correction

Line 299, “only 32.5% of the population buys herbal antimalarial drugs…(32)”: REF 32 is a study performed in Haiti. It is not relevant to the situation in Ghana. Please cite a more appropriate reference. - we have made a correction

Line 317: prevention of malaria - we have made a correction

REF 4 is incomplete: Ashley EA, Pyae Phyo A, Woodrow CJ. Malaria. Lancet. 2018;391(10130):1608-1621. doi: 10.1016/S0140-6736(18)30324-6. - we have made a correction

REF 6: Please provide more information (web link, for example). - we have made a correction

REF 7 is incomplete (page numbers): Goodman C, Brieger W, Unwin A, Mills A, Meek S, Greer G. Medicine sellers and malaria treatment in sub-Saharan Africa: what do they do and how can their practice be improved? Am J Trop Med Hyg. 2007;77(6 Suppl):203-218. - we have made a correction

REF 8 is incomplete (page numbers): Malm KL, Segbaya S, Forson I, Gakpey KD, Sampong LB, Adjei EA, Bart-Plange C. Implementation of the Affordable Medicines for Malaria Facility (AMFm) in Ghana: processes, challenges and achievements. J Public Health Policy. 2013;34(2):302-314. doi: 10.1057/jphp.2013.12. - we have made a correction

REF 9 is incomplete (page numbers): Ndeffo Mbah ML, Parikh S, Galvani AP. Comparing the impact of artemisinin-based combination therapies on malaria transmission in sub-Saharan Africa. Am J Trop Med Hyg. 2015;92(3):555-60. doi: 10.4269/ajtmh.14-0490. - we have made a correction

REF 10: Tun KM, Imwong M, Lwin KM, Win AA, Hlaing TM, Hlaing T, Lin K, Kyaw MP, Plewes K, Faiz MA, Dhorda M, Cheah PY, Pukrittayakamee S, Ashley EA, Anderson TJ, Nair S, McDew-White M, Flegg JA, Grist EP, Guerin P, Maude RJ, Smithuis F, Dondorp AM, Day NP, Nosten F, White NJ, Woodrow CJ. Spread of artemisinin-resistant Plasmodium falciparum in Myanmar: a cross-sectional survey of the K13 molecular marker. Lancet Infect Dis. 2015;15(4):415-421. doi: 10.1016/S1473-3099(15)70032-0. - we have made a correction

REF 11 seems to be wrong cited. This WHO document was not published by Trans R Soc Trop Med Hyg. World Health Organization. (‎2015)‎. Guidelines for the treatment of malaria, 3rd ed. Geneva, World Health Organization. https://apps.who.int/iris/handle/10665/162441 . Moreover, a more recent WHO document is available. Please see the web link: https://reliefweb.int/report/world/who-guidelines-malaria-3-june-2022 - we have made a correction

REF 13 is available on line. Please add the web link. - we have made a correction

REF 14 is available on line: World Health Organization. (‎2012)‎. Test, treat, track: scaling up diagnostic testing, treatment and surveillance for malaria. World Health Organization. https://apps.who.int/iris/handle/10665/337979 - we have made a correction

REF 15: Please complete this reference: page numbers or article number. - we have made a correction

REF 16: Amponsah AO, Vosper H, Marfo AF. Patient related factors affecting adherence to antimalarial medication in an urban estate in Ghana. Malar Res Treat. 2015;2015:452539. doi: 10.1155/2015/452539. - we have made a correction

REF 17: Buabeng KO, Duwiejua M, Dodoo AN, Matowe LK, Enlund H. Self-reported use of anti-malarial drugs and health facility management of malaria in Ghana. Malar J. 2007;6:85. doi: 10.1186/1475-2875-6-85. - we have made a correction

REF 18: Aborah S, Akweongo P, Adjuik M, Atinga RA, Welaga P, Adongo PB. The use of non-prescribed anti-malarial drugs for the treatment of malaria in the Bolgatanga municipality, northern Ghana. Malar J. 2013;12:266. doi: 10.1186/1475-2875-12-266. - we have made a correction

REF 19 and 20: x Ethiopia/Nigeria - we have made a correction

REF 22: Ayi I, Nonaka D, Adjovu JK, Hanafusa S, Jimba M, Bosompem KM, Mizoue T, Takeuchi T, Boakye DA, Kobayashi J. School-based participatory health education for malaria control in Ghana: engaging children as health messengers. Malar J. 2010;9:98. doi: 10.1186/1475-2875-9-98. - we have made a correction

REF 23: Please add the authors and page numbers. - we have made a correction

REF 24: x Ethiopia - we have made a correction

REF 25: Van der Wielen N, Channon AA, Falkingham J. Universal health coverage in the context of population ageing: What determines health insurance enrolment in rural Ghana? BMC Public Health. 2018;18(1):657. doi: 10.1186/s12889-018-5534-2. - we have made a correction

REF 28: Williams HA, Jones CO. A critical review of behavioral issues related to malaria control in sub-Saharan Africa: what contributions have social scientists made? Soc Sci Med. 2004;59(3):501-523. doi: 10.1016/j.socscimed.2003.11.010. - we have made a correction

REF 29: Kemppainen J, Kim-Godwin YS, Reynolds NR, Spencer VS. Beliefs about HIV disease and medication adherence in persons living with HIV/AIDS in rural southeastern North Carolina. J Assoc Nurses AIDS Care. 2008;19(2):127-136. doi: 10.1016/j.jana.2007.08.006. - we have made a correction

REF 30: Please check if REF 30 was cited in the main text.REF 30. Has been removed

REF 30: Bonful HA, Awua AK, Adjuik M, Tsekpetse D, Adanu RMK, Nortey PA, Ankomah A, Koram KA. Extent of inappropriate prescription of artemisinin and anti-malarial injections to febrile outpatients, a cross-sectional analytic survey in the Greater Accra region, Ghana. Malar J. 2019;18(1):331. doi: 10.1186/s12936-019-2967-8. - we have made a correction

REF 31: Sci 2020;2(3):49. doi: 10.3390/sci2030049 - we have made a correction

REF 32: Druetz T, Andrinopoulos K, Boulos LM, Boulos M, Noland GS, Desir L, Lemoine JF, Eisele TP. "Wherever doctors cannot reach, the sunshine can": overcoming potential barriers to malaria elimination interventions in Haiti. Malar J. 2018;17(1):393. doi: 10.1186/s12936-018-2553-5. - we have made a correction

Round 2

Reviewer 2 Report

The authors have considerably improved their manuscript and satisfactorily addressed my concerns. Many minor corrections are still necessary, most of which can be corrected by the editor. Some additional corrections given below are due to oversight during my first review.

English still needs many corrections, mostly in phrases and sentences that were added to the revised version.

MINOR SPECIFIC COMMENTS:

Lines 72-73, “Despite numerous clinical trials, no malaria vaccine is available on the market (4)”: This statement is no longer true. Since 2021, WHO recommends the RTS,S/AS01 (Mosquirix®) malaria vaccine manufactured by GlaxoSmithKline (GSK) to prevent P. falciparum malaria in children living in areas with moderate to high malaria transmission. Please see reference 1 (WHO World Malaria Report 2022).

Line 81: drug stores (instead of chemical stores)

Line 85, “campaigns organized during Affordable Medicines for Malaria Facility (AMFm)”: campaigns organized by Affordable Medicines for Malaria Facility (AMFm)

Line 94: Plasmodium falciparum in italics

Line 106, “…was made possible by made possible by rapid diagnostic test (RDT), diagnostic”: Please correct this sentence: was made possible by rapid diagnostic test (RDT).

Line 108: drug stores

Line 108, figure 1: The percentages placed over each bar can be deleted. Likewise, “%” in the left-hand column along the Y-axis (0%, 10%, 20%, 30%, 40%, 50%, 60%) can be deleted (just leave 0, 10, 20, 30, 40, 50, 60).

Lines 187-188, “p/malarial”: ??

Figure 2, graph title “Use of antimalarial drugs without medical personnel consultation”: The same title is repeated both within the graph and in the text (line 190).

Line 223, “p/malarial”: ??

Figure 3: Y-axis legend, “number of person”: Please write “number of persons”. The numbers over each bar can be deleted.

Line 230, “p/malarial”: ??

Figure 4: graph title embedded in the figure “Knows the cause of malaria”: Please delete it. The same title is given in line 233.

Figure 5: graph title “The reason of antimalarial drugs use”: Please delete it. The same/similar title is given in line 248.

Figure 6, graph title “Level of knowledge about alternative methods of malaria prevention”: Please delete it. The same title is in line 254 of the main text.

Figure 7, graph title “Level of knowledge about alternative methods of malaria prevention in relation to educational level”: Please delete it. The same title is in line 262 of the main text.

Line 278, “rapid malaria test”: rapid diagnostic test for malaria

Figure 8, graph title “why the respondents did not purchase rapid test for malaria?” Please delete it. The same title is in the text (line 287)? Please delete also the numbers of each bar. In the Y-axis legend: number of persons

Lines 293-295, “GHC”: The Ghanaian currency should be converted to USD. Most readers do not know the exchange rate between these currencies.

Line 300: did not indicate that only 24%... (?) or indicated that only 24%... (?)

Line 362-363, “The strategy of malaria treatment recommended in the WHO guidelines according to WHO guidelines”: Please correct this.

Line 437: “p/malarial”: ??

Line 494: “prevention of malaria”: Please write a complete sentence.

Author Response

Dear Reviewer

Thank you again for your insightful comments. We were happy to make corrections in the indicated places.

Your review has significantly enriched our workshop, for which we thank you.

We address each of the points in your review below.

With best regards and respect

Ewa Zieliński

English still needs many corrections, mostly in phrases and sentences that were added to the revised version. - Our text was translated by a native speaker of US nationality. Our translator claims that the English translation is correct.

MINOR SPECIFIC COMMENTS:

Lines 72-73, “Despite numerous clinical trials, no malaria vaccine is available on the market (4)”: This statement is no longer true. Since 2021, WHO recommends the RTS,S/AS01 (Mosquirix®) malaria vaccine manufactured by GlaxoSmithKline (GSK) to prevent P. falciparum malaria in children living in areas with moderate to high malaria transmission. Please see reference 1 (WHO World Malaria Report 2022). - The change was introduced in accordance with the current state and the reviewer's recommendation

Line 81: drug stores (instead of chemical stores) - The change was made in accordance with the reviewer's recommendation

Line 85, “campaigns organized during Affordable Medicines for Malaria Facility (AMFm)”: campaigns organized by Affordable Medicines for Malaria Facility (AMFm) - The change was made in accordance with the reviewer's recommendation

Line 94: Plasmodium falciparum in italics- The change was made in accordance with the reviewer's recommendation

Line 106, “…was made possible by made possible by rapid diagnostic test (RDT), diagnostic”: Please correct this sentence: was made possible by rapid diagnostic test (RDT). - The change was made in accordance with the reviewer's recommendation

Line 108: drug stores- The change was made in accordance with the reviewer's recommendation

Line 108, figure 1: The percentages placed over each bar can be deleted. Likewise, “%” in the left-hand column along the Y-axis (0%, 10%, 20%, 30%, 40%, 50%, 60%) can be deleted (just leave 0, 10, 20, 30, 40, 50, 60). - The change was made in accordance with the reviewer's recommendation

Lines 187-188, “p/malarial”: ?? - The text has been changed to Antimalarial

Figure 2, graph title “Use of antimalarial drugs without medical personnel consultation”: The same title is repeated both within the graph and in the text (line 190). - The change was made in accordance with the reviewer's recommendation

Line 223, “p/malarial”: ?? - The text has been changed to Antimalarial

Figure 3: Y-axis legend, “number of person”: Please write “number of persons”. The numbers over each bar can be deleted. - The change was made in accordance with the reviewer's recommendation 

Line 230, “p/malarial”: ?? - The text has been changed to Antimalarial

Figure 4: graph title embedded in the figure “Knows the cause of malaria”: Please delete it. The same title is given in line 233. - The change was made in accordance with the reviewer's recommendation

Figure 5: graph title “The reason of antimalarial drugs use”: Please delete it. The same/similar title is given in line 248. - The change was made in accordance with the reviewer's recommendation

Figure 6, graph title “Level of knowledge about alternative methods of malaria prevention”: Please delete it. The same title is in line 254 of the main text. - The change was made in accordance with the reviewer's recommendation

Figure 7, graph title “Level of knowledge about alternative methods of malaria prevention in relation to educational level”: Please delete it. The same title is in line 262 of the main text. - The change was made in accordance with the reviewer's recommendation

Line 278, “rapid malaria test”: rapid diagnostic test for malaria- The change was introduced in accordance with the reviewer's recommendation, the change in the indicated place and in the rest of the text

Figure 8, graph title “why the respondents did not purchase rapid test for malaria?” Please delete it. The same title is in the text (line 287)? Please delete also the numbers of each bar. In the Y-axis legend: number of persons - The change was made in accordance with the reviewer's recommendation 

Lines 293-295, “GHC”: The Ghanaian currency should be converted to USD. Most readers do not know the exchange rate between these currencies. - The change was introduced in accordance with the reviewer's recommendation, the change in the indicated place and in the rest of the text

Line 300: did not indicate that only 24%... (?) or indicated that only 24%... (?) -The change was made in accordance with the reviewer's recommendation

Line 362-363, “The strategy of malaria treatment recommended in the WHO guidelines according to WHO guidelines”: Please correct this. -The change was made in accordance with the reviewer's recommendation

Line 437: “p/malarial”: ?? - The text has been changed to Antimalarial

Line 494: “prevention of malaria”: Please write a complete sentence. - The text has been removed